# Inequalities in Academic Work during COVID-19: The Intersection of Gender, Class, and Individuals' Life-Course Stage

Anna Carreri [1,2,*], Manuela Naldini [3] and Alessia Tuselli [4]

1    Department of Human Sciences, University of Verona, 37129 Verona, Italy
2    School of Social Sciences, Hasselt University, 3500 Hasselt, Belgium
3    Department of Cultures, Politics and Society, University of Turin, 10124 Turin, Italy; manuela.naldini@unito.it
4    Department of Sociology and Social Research, University of Trento, 38122 Trento, Italy; alessia.tuselli@unitn.it
*    Correspondence: anna.carreri@univr.it

**Abstract:** Research studies on academic work and the COVID-19 crisis have clearly shown that the pandemic crisis contributed to exacerbating pre-existing gender gaps. Although the research has been extensive in this regard, it has focused more on the widening of the "motherhood penalty", while other groups of academics are blurred. Even more underinvestigated and not yet fully explained are the intersections between further axes of diversity, often because the research conducted during the pandemic was based on a small volume of in-depth data. By drawing on interview data from a wider national research project, this article aims to contribute to this debate by adopting an intersectional approach. In investigating daily working life and work–life balance during the pandemic of a highly heterogeneous sample of 127 Italian academics, this article sheds light on how gender combines with other axes of asymmetry, particularly class (precarious versus stable and prestigious career positions) and age (individuals' life-course stage), to produce specific conditions of interrelated (dis)advantage for some academics. The analysis reveals three household and family life course types that embody the interlocking of gender, class, and age within a specific social location with unequal, and possibly long-term, consequences for the quality of working life, well-being, and careers of academics, living alone or with parents, couples without children or with grown-up children, and couples with young children and other family members in need of care.

**Keywords:** academic work; academic career; COVID-19 pandemic; gender; intersectionality; work–life balance; class





## 1. Introduction

Despite the increase in the number of women in universities, especially at the early stages of education, significant inequalities persist as academic trajectories progress. In general, studies have revealed that women progress more slowly up the academic ladder, they are more in charge of the less rewarded academic chores, they tend not to easily attain leadership roles, and they earn less than men in comparable positions (Cois et al. 2023; Heijstra et al. 2017; Peterson 2016; Van den Brink and Benschop 2012, 2014). This gender disparity is often captured through three metaphors: the leaky pipeline metaphor (women are more likely than men to drop out of careers), the glass door metaphor (women are less likely to gain access to stable positions) (Picardi 2019), and the glass ceiling metaphor (women are less likely to gain access to top positions). How, in this scenario, did the pandemic crisis contribute to changing academic labor and to modifying pre-existing gender gaps? A huge body of scientific research has addressed this question and shows that COVID-19 contributed to exacerbating pre-existing gender gaps, especially for those in the early stages of their careers and in temporary research positions (King and Frederickson 2021; Douglas et al. 2022; Pereira 2021; Docka-Filipek and Stone 2021; Squazzoni et al. 2021) with possible long-term consequences in terms of career entry and advancement. However, research on gender disparity in academia has not yet explained the mechanisms

and processes behind this widening gender gap and its intertwining with other axes of diversity, such as class (precarious versus stable and prestigious career positions) and age as two highly gendered social phenomena.

This article aims to contribute to this debate by analyzing daily working life and the work–life balance of both early and advanced career academics during the pandemic in Italy through an intersectional lens (Crenshaw 1989). Specifically, it investigates how gender intertwines with other axes of asymmetry, i.e., class and age, to produce specific conditions of (dis)advantage for some academics.

In the following, after a reconstruction of the State of the Art of recent but burgeoning debates about academic work and gender inequalities during the pandemic (Section 2), we present our intersectional approach (Section 3). Section 4 outlines the empirical research by describing the research context, the data, and the type of analysis we conducted. Section 5 summarizes the results of the analysis through an intersectional lens, which looks at the multiplicative effect in the intersections of different axes within a specific social location. In Section 6 we discuss the results; finally, in the last section, we offer some concluding reflections.

## 2. Gender Inequalities in Academic Labor during COVID-19: State-of-the-Art Research

Research studies on the impact of the pandemic on academic work and the consequences for careers that account for the gendered and gendering effects of COVID-19 are numerous. The debate has been extensive and is clearly ongoing. Here, we try to summarize some of the main results of the research carried out on the topic and distinguish between them according to the specific topic of investigation.

The first object of inquiry was the impact of the emergency on the time devoted to work rather than to other activities and times of life with comparisons between the experiences of male and female academics (Caldarulo et al. 2022; Derndorfer et al. 2021; Ghislieri et al. 2022; Myers et al. 2020; Yildirim and Eslen-Ziya 2021). These studies show that, among the various components of academic work, time devoted to research was most affected by the pandemic, decreasing by an average of 24% (Myers et al. 2020). However, there are important differences. For example, among the various disciplines, those involving the use of laboratories suffered the most significant setbacks, while regarding gender, the most severe impact was on women and particularly, mothers (Derndorfer et al. 2021; Ghislieri et al. 2022; Myers et al. 2020; Yildirim and Eslen-Ziya 2021).

The second object of inquiry was how remote work (so-called "home-working") was experienced during university lockdowns from a subjective point of view. These studies make it possible to explore the quality of working life. They show how experiences differ greatly not only between women and men but also according to family configuration and the stage reached in an academic career. For example, the research results suggest that academics in the most precarious positions are those who have experienced particularly stressful conditions under the imperative of maximum productivity, with harmful effects on their psycho–physical well-being (Carreri and Dordoni 2020; Docka-Filipek and Stone 2021; Douglas et al. 2022; Hadjisolomou et al. 2022), albeit with variable impact according to the characteristics of the employment contract, the existence of teaching duties, the support of a mentor, cohabitation with others, and belonging to a minority group in terms of nationality, disability, or parental responsibility (Carreri et al. 2023; Douglas et al. 2022; Kınıkoğlu and Can 2021). These studies also show that the pandemic strongly compromised the capacity to concentrate on work, as well as the mental health of women and even more so of mothers (Carreri and Dordoni 2020; Docka-Filipek and Stone 2021; Górska et al. 2021). Furthermore, these studies shed light on a part of academic work, mostly carried out by women, which often remains invisible and which is neither recognized nor valued, for instance, mentoring and tutoring students, activities related to course preparation or services, or other time-consuming activities that increased exponentially during the pandemic and were made more complicated by remote home-working (Carreri et al. 2023; Górska et al. 2021). Many women, therefore, saw the time dedicated to their double role of

care—both in the family and at university—increase during the pandemic (Altan-Olcay and Bergeron 2022; Docka-Filipek and Stone 2021; França et al. 2023).

The third widely explored theme was the impact of the pandemic on scientific productivity, which is a key factor in building an academic career. The studies highlight how during the first phase of the pandemic, for all disciplines, productivity (measured in terms of number of submissions and publications) decreased more for women compared to previous years or did not grow as much as it did for men, especially for women with children (Amano-Patiño et al. 2020; Andersen et al. 2020; Cui et al. 2021; Kibbe 2020; King and Frederickson 2021; Flaherty 2020; Squazzoni et al. 2021; Viglione 2020) and for women in the first stages of their academic careers (Amano-Patiño et al. 2020; Andersen et al. 2020; Vincent-Lamarre et al. 2020). A similar trend is apparent regarding the presentation of projects and fundraising activities (Gao et al. 2021; Kalpazidou Schmidt 2020). Conversely, some academic service activities that are undervalued but fundamental for the functioning of the scientific community, such as peer reviews, grew more for women than men (Squazzoni et al. 2021).

Also, based on these results, a distinct stream of studies developed focusing on what was known well before the pandemic as the "motherhood penalty"[1] (Lutter and Schröder 2019) when the experience of being an academic mother is compared to men and to women without care responsibilities (Kasymova et al. 2021; Minello 2020; Minello et al. 2021). The disadvantage to mothers (with small children) who work in universities is generally attributed to two factors: the first concerns the gender division of unpaid work within the home; and the second concerns the gender division of academic work mentioned above (Minello 2021). During the pandemic, mothers often had to postpone (until nighttime) or abandon more intellectual (and conceptual) work related to scientific research so that they could devote themselves (in addition to domestic work) to childcare, as well as to academic work but mostly in the form of teaching, student tutoring, and service activities. Furthermore, around this theme, a space has developed for the sharing of subjective experiences, for joint reflection, and for claiming a set of necessary changes within academic contexts (see, for example, Boncori 2020; Bowyer et al. 2021; Couch et al. 2021; Guy and Arthur 2020; Heath et al. 2022; Plotnikov et al. 2020). This is a space of auto-ethnographic and, in many cases, creative writing by academic mothers, which yields a detailed account of the micropolitics driving the reproduction of gender inequalities and sheds light on mechanisms that are often pre-existing but are not discussed in the typically male academic context described as a "care-free zone" (Lynch 2010).

Importantly, there is a final issue discussed in the scientific literature that concerns the policies and actions (not) adopted by universities, including those necessary to adequately respond to the disadvantages that the pandemic exacerbated for some groups, especially academic mothers (Gewin 2020; Malisch et al. 2020; Nash and Churchill 2020; Oleschuk 2020; Mickey et al. 2023). These studies, although small in number, highlight how universities, in such extraordinary times, often shirked their responsibility to ensure that academic staff with domestic and care tasks—largely female—could fully participate in academic work, interpreting work–family balance as an exclusively "private" matter (Nash and Churchill 2020).

It is notable how these studies invite us to shift our perspective and not only consider the disadvantages of mothers but also reflect on the choice to have, and especially not to have, children made by women wanting to pursue an academic career—especially if they have nontenured posts (Naldini et al. 2023)—and, therefore, to consider the existence of an implicit selection process (Gaiaschi 2022; Minello 2021). On the other hand, studies (implicitly) risk making all fathers fall within the model of "hegemonic masculinity" (França et al. 2023) and validating the assumption that the obstacle against achieving the "right" level of productivity—"business as usual"—in times of crisis consists (exclusively) of the care responsibilities that fall mainly on women (Corbera et al. 2020; Pereira 2021; Utoft 2020), pushing the role of family configurations and the partner's employment conditions into the background (Martucci 2023).

### 3. Intersectional Lenses and an Intracategorical Approach to Academic Work: Perspective and Categories

The intersectional perspective (Crenshaw 1989) assumes that people have multiple and layered identities resulting from social relations, history, and power structures. Adopting this perspective allows us to grasp complexity by investigating how various factors (gender, race/ethnicity, class, sexuality, religion, and others) combine and create specific forms of interrelated discrimination. The intersectional perspective is the key to considering how various identity factors are interconnected on both individual and institutional/social/cultural levels (micro and macro levels) (Castro and Holvino 2016). Intersectionality underlines the complex inequalities generated within power relations, going beyond the analysis of any single category or even the mere sum of several categories (Cho et al. 2013; Collins 2000, 2015; Crenshaw 1989).

Intersectionality was theorized in the early 1990s by combining gender and racial factors (Crenshaw 1989), but since then, scholars have become more and more aware of the fact that two categories may not be enough to tackle social inequalities in societies (Lutz 2023). Many intersectionality scholars have advocated for the incorporation of other categories, such as nationality (Anthias and Yuval-Davis 1992), sexuality (McClintock 1995), and other markers of discrimination, such as class, age/generation, health, disability, and gender identity (Amelina and Lutz 2019). It is important to consider that identity categories are nonadditive and may acquire a (slightly) different meaning in different contexts. Which axes, demographic characteristics, or inequalities should be considered to select "particular social groups at neglected points of intersection" (McCall 2005, p. 1773) is, indeed, still the subject of academic debate.

Today, intersectionality is a well-established approach in feminist studies (Davis 2008; McCall 2005), but it is a concept that has gone far beyond gender studies and is adopted in many disciplinary fields of social sciences. For example, recent studies on organization have begun to adopt an intersectional perspective (Banerjee 2012; Holvino 2010; McBride et al. 2015) with the aim of analyzing the ways in which gender relates to other sources of inequality (Muzio and Tomlinson 2012). It can be said that intersectionality is, itself, the subject of reinterpretation, gradually including categories not initially considered, and has become a "buzzword": indeed, the term has the advantage of being open-ended and providing adaptability to diverse contexts (Davis 2008).

Intersectionality as a concept and analytical tool has generated heated debates within different disciplinary fields. The focus of intersectional approaches has also shifted to other "intersections", giving rise to new theories and holding together a variety of ways of understanding and approaching intersectionality (Cho et al. 2013). In this regard, it is useful to mention the distinction proposed by Nina Lykke (2010, spec. pp. 68–69) between (i) "explicit" theories of intersectionality that explicitly use the term introduced by Crenshaw; (ii) "implicit" theories of intersectionality that have explored the intersections of gender, sexuality, race, and class, without the direct use of the word "intentionality" (including, for example, Black Feminism studies before the term was introduced); and (iii) theories of the intra-actions of social categories/structures that propose "other designations" to analyze complexity, including "interlocking systems of oppression" (Combahee River Collective 1977; Collins 1991), "axes of power" (Nira Yuval-Davis 2006), "interferences" (Moser 2006; Geerts and Van der Tuin 2013), "cosynthesis" (Kwan 2000), and "interdependencies" (Hornscheidt 2007).

Within this articulated methodological debate, Leslie McCall (2005) elaborates on the first method to apply intersectional theory in research. McCall (2005) believes that intersectionality represents "the relationships among multiple dimensions and modalities of social relations and subject formations" (p. 1771), and in the article "The Complexity of Intersectionality", she describes three possible approaches on how to use the intersectional lens. The different approaches depend on how social categories are treated: anticategorical, intracategorical, and intercategorical.

Studies on "anti-categorical complexity" refer to the deconstruction of categories "as part and parcel of the deconstruction of inequality itself" (McCall 2005, p. 1777). According to this approach, categories are the result of an arbitrary social construction and do not represent the complexity of subjects and social structures. The proposal is to overcome the use of categories to target those social constructions that generate inequality (McCall 2005). McCall underlines that the anticategorical approach makes it possible to challenge a system of thought that considers identity categories in an univocal and rigid manner; at the same time, she recognizes the importance of using categories because the process of decategorization brings with it "political consequences" (Matsuda 1990, p. 1776). Categories make visible the complexity of social inequalities, and they emphasize the presence or absence of privileges with respect to their positioning in society.

Therefore, McCall proposes the second so-called intracategorical methodological approach: studies that critically use categories to investigate "minorities within minorities" (McCall 2005, p. 1780). The focus is on the production and reproduction of categories in social life to reveal the complexity of the experience of discrimination lived within each social group (McCall 2005). Intracategorical analysis examines groups or individuals, considering as many categories as possible (McCall 2005). McCall identifies qualitative research as the way to analyze individual specificity, particularly in case studies and semistructured interviews. Qualitative methods generally have the capability to investigate the complexity of society in order to represent its diversity and heterogeneity (Ragin 2000).

The last intercategorical approach focuses on "the nature of the relationships among social groups and, importantly, how they are changing" (McCall 2005, p. 1785). The aim is to consider the relationships between social groups and define what McCall calls the "configurations of inequality" (McCall 2005, p. 1789). The emphasis is on the transformations that occur within the relationships between groups rather than on the individual level. The corresponding methods are those of quantitative research: an empirical analysis of the multiple dimensions in which social categories are shaped in a comparative sense.

McCall's model takes up the challenge of "intersectionality, yes but how?" (Hvenegård Lassen and Staunæs 2020). Systematizing a complex approach such as intersectionality is difficult, and although models can be useful in many ways, they can also oversimplify (cf. Lutz and Amelina 2021b). With reference to McCall's proposal, each approach has its limitations and opportunities: the intercategorical approach takes into account three social categories—gender, class, and race—and explores their mutual interpenetration, but the categories are defined as "natural" and static; the intracategorical approach, on the other hand, takes into account the transformation processes of the categories, which are situated and change in space and time; and the anticategorical approach challenges the identity categories themselves, emphasizing the social constructions and simplifications that may result from them. Notwithstanding the potential and criticalities present in each approach, it is important to consider that it is difficult to separate the approaches clearly in a research process (Amelina 2021; Lutz and Amelina 2021a, 2021b), as well as to place oneself clearly and unequivocally within one of the three.

For this research, embracing an intersectional perspective means observing how gender intersects with other social categories to produce specific conditions of (dis)advantage for different groups within academia. Therefore, we will adopt an intracategorical approach while taking into account the methodological reflections made so far. Indeed, in this study, we consider how, during the COVID-19 pandemic, the combination of gender, age, and class (career position) produced specific asymmetries in the academic context by bringing out the relationships between individual experiences, organizational cultures, and certain dimensions related to parenting models. Indeed, the choice of the three axes is connected to the existence of peculiar power relations in the academic field and how these intertwine with individual experiences. In academia, both in Italy and in Europe, significant gender disparities remain. Women have greater difficulties in career paths (European Commission 2021), in gaining access to certain courses of study (such as STEM studies), and in being represented within academic governance (Checchi et al. 2018; Murgia

and Poggio 2019). Therefore, it is crucial to make these processes visible through the lens of gender studies and to remember that gender combines with other identity factors in our specific research context, such as age. Age is defined here as the "individual life-course stage" and is the second axis of intersection that will be analyzed. Today, younger people working in Italian universities occupy precarious positions for longer than previous generations, and they are evaluated through a system based on performance and entrepreneurship (Gaiaschi 2021). At the same time, they are at a time of their life that may coincide with their desire for a family, which is often difficult to combine with job insecurity and new standards of academic productivity. The interaction between gender and the life-course stage gives rise to specific forms of asymmetry, which become more complex if class, understood as academic position, is also taken into account: occupying stable or precarious positions offers very different life prospects, possibilities, economic means, (in)stability, and (in)security, as we observe below. Recent reforms in the Italian university context have facilitated a process of the precarization of scientific careers, also driven by the frequent use of fixed-term contract forms and the associated job instability and by the continuing uncertainty in access to economic resources (Bellè et al. 2015). In Italy, those in precarious employment positions with a fixed-term contract may find themselves excluded from forms of protection and welfare and income support policies. In the European context, this situation is only present in Italy. "The absence of income support and/or continuity of income itself has the effect of reproducing inequalities based on social class of origin and/or the availability of one's own or one's family's resources, to be deployed during periods when one continues to do research even without a salary" (ibid., p. 68). A category of precarious researchers is also called the working poor, who have no or few protections in the present as well as in the future (if one thinks of retirement) and who find themselves in a very different position than the professors with whom they have a collaborative relationship, in terms of class, income, and life perspective (Coin et al. 2017).

## 4. Context, Data, and Method

### 4.1. The Italian Academic Context

In Italy, the problem of gender inequality in academia becomes evident after Ph.D. graduation, in the early stages of scientific careers (leaky pipeline and glass door phenomena), and much more so in the later stages (glass ceiling phenomenon). Since the mid-2000s, to address a serious financial crisis, Italy has progressively introduced reforms inspired by the New Public Management paradigm (Krüger et al. 2018). The Italian academic context has undergone profound changes and financial shortfalls as a result of three policies: the adoption of a new evaluation system for departments and universities, the so-called Gelmini Reform (Law No. 133/2008 and Law No. 169/2008), and cuts in public funds for universities and research, which have also affected academic staff turnover (Gaiaschi and Musumeci 2020). The first policy created the national evaluation system (VQR—Research Quality Assessment; "Departments of Excellence"). Currently, how the MIUR allocates funding to universities partially depends on the result of this evaluation (Gaiaschi and Musumeci 2020). The second reform enacted by Law No. 133/2008 concerned, in particular, the early stages of an academic career. The Law introduced two different kinds of fixed-term contracts: researcher A (RTDa: fixed-term researcher) and researcher B (RTDb: permanent researcher), replacing the indefinite-term researcher (RU) contract. This started a process of precarization of the early stages of the academic career path (Bozzon et al. 2015). The last measure reduced turnover in universities (which had characterized the period between 2007 and 2017) by not allowing universities to proceed autonomously with staff recruitment. This led to no replacement of retired lecturers with newly hired personnel (Gaiaschi and Musumeci 2020) and contributed to making university careers more difficult and slower (Guarnascio et al. 2023). Recent studies (Gaiaschi et al. 2018; Picardi 2019) underline how the combination of these three factors has had repercussions from a gender perspective, particularly in the selection processes in the early stages of the academic career path, especially for fixed-term researcher positions.

These transformations are part of a framework in which Italian universities are increasingly embracing a neoliberal conception of scientific productivity and its evaluation through certain parameters, such as competition (among researchers, departments, and universities) and accountability (Gaiaschi and Musumeci 2020; Poggio 2018b; De Coster and Zanoni 2019). It is important to note that this system is specific to Italian academia and is compounded by the fact that there is significant mobility between academic institutions within Italy.

### 4.2. Data and Method

Against this backdrop, the national research project GEA—Gendering Academia, which was funded by the Italian Ministry of Universities,[2] was aimed at investigating, from a gender perspective, the academic careers and working conditions of women and men doing research and teaching work in different career stages and disciplines in four Italian universities, with a focus on recent transformations in the university and the processes of recruitment. This article draws on qualitative data collected in GEA—Gendering Academia's project. More specifically, it is based on the analysis of 127 qualitative interviews with two groups of academics at different levels of their academic careers but who are not full professors (yet): postdocs and temporary researchers (Early Career academics (ECas)) and associate professors (Advanced Career academics (ACas)), working in both STEM and SSH academic departments in four Italian universities located in different parts of the country (two from the North and two from the South).

The sample was identified by its theoretical significance and included both women and men, both with children and without, for each department, disciplinary area, and career level, with the goal of maintaining heterogeneity in terms of age and academic seniority and including different career patterns of both men and women.

The ECa sample consisted of 32 men, half in SSH and half in STEM, and 32 women, including 11 fathers and 10 mothers, while the others had no children. The ages ranged from 27 to 46 (mean age 35). At the time of the interview, 30 participants were fixed-term researchers, 32 were research grant holders, and two were research fellowship holders and contract lecturers. (Some ECas also had contract lectureships.)

The interviews with ACas involved 31 female associate professors, 16 in SSH and 15 in STEM, and 32 male associate professors equally distributed between the two fields of study, aged 40 to 66 (with an average age of 49), including 19 women and as many men with children. The online semistructured qualitative interviews were conducted in the four Italian universities during the COVID-19 pandemic crisis. Specifically, the interviews, which, on average, lasted 1 h and a half, were conducted online between May 2020 and May 2021, a year characterized in Italy not only by the first severe lockdown, which lasted almost two months, but also by various restrictive measures.[3] In March 2020, Italy was the first country in Europe to impose a nationwide COVID-19 lockdown. The law imposed an immediate suspension of all commercial and industrial activities, with the exception of "essential" sectors. These were economic activities that were considered essential to sustain citizens and support the economy during the pandemic. All businesses and institutions in other sectors could only remain active via remote working. Therefore, sports and cultural events were suspended, and schools and universities were closed. In addition, the government imposed travel stoppages and strong limitations on personal mobility. Several lockdowns were imposed, and in the fall of 2020, the management of the pandemic with its restrictions was differentiated from region to region according to the incidence of cases. In this context, remote work and distance learning remained a widespread form of work and organization in university. In spring 2021, vaccines became available, and the restrictions eased as a result.

The interview outline contained six sections: individual academic career, current daily working life and pandemic changes, organizational cultures (current and past), well-being and work–life balance and pandemic changes, and perceptions of and satisfaction with their current position, future prospects, and policies. After having fully transcribed and

anonymized the interviews (all names are pseudonyms), a thematic analysis was carried out in several steps following an iterative process, with the support of the qualitative analysis software Atlas.ti 9. The analysis explored in depth the daily work life and work–life balance of both early and advanced career academics. In a second more interpretative phase, we sought to shed light on the intersections of three axes of diversity—gender, social class, and age—which were made more visible by the pandemic. In this phase, we identified three "conformations" that best embody the intersectionality of these axes as they show specific outcomes in terms of quality of working life, well-being, and potentially unequal long-term consequences for the careers of academics. Throughout the analysis process, the codes and their interrelations were discussed among the authors, and a continuous conversation was maintained between the coding and theoretical interpretations. The analysis revealed three household and family life course types that highlight the interlocking of the three axes producing specific conditions of (dis)advantage for some academics: living alone or with parents (32 interviewees), couples without children or with grown-up children, i.e., over 14 years old (52 interviewees), and couples with young children (under 14 years old) and other family members in need of care (43 interviewees).

In terms of disciplinary areas, both STEM and SSH staff started using more individualized ways of working, albeit with some differences. Those working in STEM had to interrupt their team experiments in the labs but without slowing down their publication activity. In fact, STEM staff used the time freed up by the lack of lab activity to develop papers, read data, and write new articles. With the reopening of labs in the later stages of the emergency, a new, precise division of tasks was developed that had to be performed individually to comply with the new health regulations and social distancing. In the SSH field, on the other hand, research activity was characterized by the fact that it was mainly carried out individually, even before the emergency, and what the interviewees lacked most was the exchange of ideas—even informally—between colleagues that normally took place in the organizations. Importantly, this discipline-based characterization does not differ between the three household and family life types we focus on below.

## 5. The Pandemic and the Interlocking of Gender, Class, and Age

### 5.1. Living Alone or with Parents during the Pandemic

Those interviewees who were at a stage in their life course with neither partner nor caregiving burdens for young children or older adults enacted an "*inhuman*" management model, where there was a (forced, in some respects) withdrawal from the sphere of private and social life. As Pino recounted, people entered "*another dimension*" that seemed suspended in time, alienated from the world, and in which social relations had been compromised.

> Social life has been greatly compromised, uhm… […] so we are in a time in some ways of… of transition which has been going on for several months now, in which I must say we are also beginning to forget the type of life that we previously had […] Now [during the pandemic] we've suddenly entered another dimension. So, more than anything else the work dimension has now shifted into private spaces, but it occupies spaces that in fact could not be devoted to anything else, if not the family dimension… All other relationships have been somehow compromised. (Pino, 56 years old, SSH field, ACa.)

In particular, during lockdowns, the pandemic seems to have generated for interviewees who lived alone or with their parents what were called "*empty times*" that were no longer devoted to social/private life and had to be "*filled*". Often, in these cases, regardless of gender, the strategy of overwork was adopted, in line with the organizational academic culture (Cannito et al. 2023), with the consequence of a complete invasion by work of what had become a single productive time–place.

> It was a tragedy [laughing] in the sense, I probably worked twice as hard, especially in the initial phase of the lockdown, but trivially because I didn't have

anything else to do. . . So my time management changed because I could work calmly. I woke up later because I didn't have to catch the bus, and so on. I drank coffee, switched on the computer and started working at nine and finished at eleven in the evening, quietly. (Enzo, 31 years old, STEM field, ECa.)

With the pandemic [time management] got worse because. . . obviously not being there, by anyone. . . then the days were all the same so I carried on working like this for weeks. (Erika, 40 years old, STEM field, ACa.)

In terms of social class, we note that this complete invasion by work and withdrawal of time for oneself takes on a different character in the experiences of early and precarious career academics and those in stable positions. Specifically, for the latter, the increase in work activities especially concerned highly demanding experimental distance teaching and more frequent online meetings related to governance, services, and "care" activities, which were more thematized by women, as shown in other research (Altan-Olcay and Bergeron 2022; Docka-Filipek and Stone 2021), with a consequent reduction of the time spent on research.

Importantly, the pandemic was described by several interviewees as a positive period to the extent that it allowed academics to "*gain time*" and be "*more productive*", demonstrating the extent to which the model of the "ideal academic" (Cannito et al. 2023; Lund 2015) was introjected. The "ideal academic" must be assertive, ambitious, and able to survive competition, characteristics traditionally associated with the hegemonic model of masculinity. In addition, the "ideal academic" is expected not to interrupt or slow down work commitments. Slowdowns are not only not contemplated, but when they do occur, they result in both reputational and career penalties (Cannito et al. 2023). In terms of social class, especially for those in early and precarious posts, working from home during the lockdowns freed up time for writing articles and projects, making it possible to enhance the components of academic work that are most valued individually—in terms of funds, prestige, and career entry—by neoliberal organizational cultures (Poggio 2018a) as well as Italian national scientific qualifications. While, on the one hand, the pandemic imposed various constraints on the performance of academic work (in terms, for example, of building important informal networks for ECas and teaching and building good relations with students, especially felt by ACas), on the other, paradoxically, it made it possible to deploy the model of the "ideal academic" more openly (Lund 2015). Furthermore, the analysis shows that, in the case of academics at the beginning of their careers and in particular for those with research grants, new teaching and tutoring activities that should have started were suspended or canceled due to the health emergency. Significantly, several respondents said they were relieved by the lack of these opportunities because the remote work would have taken up too much of their time and distracted them from their primary research objectives.

This also has, let's say, a positive aspect. . . because the more time you have available, the more publications you can develop: therefore, basically. . . you are then able to pay more attention to publications, to take care of that phase. (Nicola, 35 years old, SSH field, ECa.)

I had more time, it was great [laughs], I wrote more, I was more productive, absolutely positive. (Lia, 29 years old, SSH field, ECa.)

The "saving" of time (the "gain") was clearly felt by those interviewees who had interrupted their lives as commuters and by those who had suspended several personal activities, such as sport.

Let's say that from this point of view things have improved, because being at home has made it possible to really optimize my time, because. . . Just think that I had to go to **** and stay there a week for lectures, but during this time I did them [lessons] from the comfort of my home. . . So from that point of view, Covid relieved me because I did all the lessons online then. (Barbara, 46 years old, SSH field, ECa.)

Interestingly, the analysis did not reveal significant differences between men and women. However, in focusing analytical attention on inequalities, not to be forgotten are those that were implicit in the texts collected and were socially invisible. Importantly, the pandemic, by giving unprecedented visibility to existing asymmetries and exacerbating them, caused some female interviewees, particularly those in terms of the class with nontenured and precarious posts, to reflect on whether to have and, above all, not to have a family in order to be able to continue their academic career. However, it was a nonchoice—i.e., the inability to imagine a family for oneself—as can be read in the next excerpt.

> I was just talking these days with some of my colleagues who have families, and it was devastating. It has been devastating to have to go home with Covid, because they are in fact the ones who have to carry the load... So I feel privileged precisely for not having to carry that load on top of what I already do. It's probably also a dog biting its own tail, in the sense, I can't imagine having a family in the immediate future, because I realize that there is no room in my life for even a seedling. (Giusy, 29 years old, SSH field, ECa.)

*5.2. Living in a Couple without Children or with Grown-Up Children*

As in the case of the previous household type, the interviews with those who were at a stage in their life course with a cohabiting partner but without caregiving burdens showed a strong culture of hyperwork, without distinction of gender and both in the STEM and in SSH. Compared to those living alone, however, we did not find a feeling of being completely absorbed in the work sphere and alienated from the social world. The interviewees worked long hours, but they did so while reconciling this with their private lives. As Francesca, an associate professor in the STEM field, said: "*We worked a lot, but we worked well*". This was especially true of those whose family circumstances allowed them to have a good degree of control over their space–time: firstly, those who had no one that partly or wholly depended on them; secondly, the members of couples who worked at university or professionals who worked remotely in a spacious house, a condition that advanced career academics, in particular, were able to take advantage of as it is related to social class. Associate professors, being financially well off and having large houses with several rooms to work in, represented a particularly advantaged group in our sample. In the experience of Harry, an associate professor with two grown-up children, with the pandemic, the management of workplace–times was "*easier, much easier: just change rooms and change clothes*". Likewise, Ignazio, a member of a childfree couple, said:

> Fortunately, we could give lectures in two separate rooms, and from this point of view we didn't get in each other's way... Living as a couple, each of us with a computer in two different rooms, we could... we had the space. (Ignazio, 58 years old, STEM field, ACa.)

Surprisingly, the lockdown also had positive effects on private and family life. On the personal side, the period of confinement was seen by some interviewees as beneficial because it freed up time and expanded "space for yourself", especially for early career academics without teaching duties. For example, Patrizia, an early careerist in the STEM area, said that with the closure of the laboratories she had rediscovered her personal life.

> Paradoxically, with the pandemic I've rediscovered, let's say, I haven't rediscovered but I've managed to carve out, some space for myself, for my personal/private life... which I wasn't able to do before, I wasn't able to have, because spending the whole day in the laboratory I couldn't carve out that space for myself, because even if you take a break in the laboratory you still don't do, you don't do activities that aren't part to your professional activity. (Patrizia, 34 years old, STEM field, ECa.)

As regards family life, for several interviewees, the lockdown resulted in a pleasant (re)cohabitation. For example, Carmine, an early careerist with a child on the way, said that, for him, COVID-19 was "a godsend" that had brought him a better quality of life–work

conciliation because he could share the months of his partner's first pregnancy. On the other hand, Harry, an advanced careerist, said that he had "rediscovered his children" during the lockdown:

> Well, I've rediscovered my children, who I used to see very little. Among other things my daughter had just returned from an Erasmus exchange in Paris, and I hadn't seen her for six months. I mean, I'd gone to see her once. Then… we never said much at home, for so many years, we'd all been at home… Luckily we have a big house so there was no… A very connected big house and we have several computers, so there wasn't even the problem of say everybody in the same room and… everybody without connection or anything like that. It was never a problem thankfully. (Harry, 55 years old, SSH field, ACa.)

Positive experiences were also recounted by all the couples without children and with additional housing and online resources, a socially class-dependent condition. This was the case for Stella, who had decided to spend the lockdown period in her parents' house outside the city. When asked about the repercussions of COVID-19 on her work–life balance, she replied:

> No, it's gone well. We've moved in with my parents, who have a house in the countryside. So the pandemic immediately became more liveable, and I have a dedicated room as if it were my studio. I have everything I need to work, a big screen, a printer… For me it's been easier compared to staying at home, let's say. The four of us divide the work in a different way and then we keep each other company. (Stella, 48 years old, SSH field, ACa.)

It was observed that women were more likely than men to thematize an activity and participate in mental reflection, which was necessary to redefine the boundaries between professional life and private life, as shown in other studies (Carreri and Dordoni 2020). Moreover, domestic work—which, for many interviewees, especially of a higher social class, had been outsourced before the pandemic—ended up falling disproportionately on the shoulders of women, thus generating the "double presence" described during the 1970s by Balbo (1978), and which is still a very topical and useful metaphor with which to represent the unprecedented overlap between work and the private/family sphere. However, it emerged from the analysis that for women who now had grown-up children and more "egalitarian" husbands, the reorganization of routine had not produced a further imbalance in life–work conciliation compared to the previous period. For example, Lena, an advanced career academic with two older but not yet adult children, had been forced by the lockdown to reduce the time that she allocated to rest and leisure because, unlike those at an early career stage, she had many online teaching commitments and meetings, but thanks to her husband's support, she had not seen an increase in domestic and care work.

> Yes, [the lockdown] has greatly reduced rest times, because as I said, online teaching, especially in the past semester, has absorbed me a great deal… However, our home has been reorganized, in the sense that, because my husband is at home, many of the care tasks and cleaning chores, and so on, have been taken on by him. I must say it has obviously had an impact on the overall management… It hasn't had a heavy psychological impact, not nearly as heavy as in cases with younger children, in smaller homes. But for rest time… and free time, too, they have closed the gym to me [said ironically]. (Lena, 51 years old, SSH field, ACa.)

### 5.3. Living with Young Children and Other Family Members in Need of Care

Those interviewees who lived with young children encountered greater reconciliation difficulties in home-working during lockdown. The closure of services and schools meant, especially for mothers, a complete overlap between work and the management of everyday life, without it being possible to count on grandparents, who are an essential resource in the Italian welfare system for the care of children (especially if young and of school age).

As underlined by some research studies on how the children and family care burden was redistributed during the pandemic (Del Boca et al. 2020; Kulic et al. 2020; Naldini 2021), our analysis also highlighted logistical, physical, emotional, and intellectual difficulties in balancing work and family; difficulties perceived as more penalizing in terms of career for women compared to their male colleagues in the same professional position and with the same family circumstances.

> I've realized in this Covid period what it means to be a woman or to be a man in a job like mine. I've seen my male colleagues who were delighted to give their lectures online because they could stay there [home], do their stuff, without anyone hassling them. But I, on the other hand, was lost behind my daughter's home-schooling. So. . . I'd planned these spring months to do a whole series of things. Instead, I'm here [at home] and still have to try to manage these things. (Elvira, 51 years old, STEM field, ECa.)

The difficulties mainly affected mothers with several young children. For example, Nora, a mother of three children (11, 9, and 3 years old), represented the different day at Mom and Dad's house:

> Now, I have to try to carve out time and devote it to work with a basis of concentration where every ten minutes someone comes and asks me something, including guilt at the time when one treats one's children badly because they were trying to focus on something, and you have to live with that I think. . . The change is radical on moms, on women, because a dad who works from home stays locked in his room and works; the mom who works from home is not like that. . . So the gender difference is there objectively. (Nora, 38 years old, STEM field, ECa.)

They felt tired, could not sleep, and experienced stress and, sometimes, confusion about space–time boundaries (Carreri and Dordoni 2020). They said that they "produced" (published) less or nothing, and when they were able to do so, it was with extreme suffering and, in the long run, with possible effects on their careers. Hence, a gap—that of the "motherhood penalty"—opened up within the same group of women, between those who were childfree (and adhered in some way to the neoliberal model) and those who had care responsibilities (and tried to embrace the neoliberal model but in a partial way). Like the findings in other studies (Kasymova et al. 2021), this gap is a subject of reflection only for mothers and some childless women.

Importantly, the pandemic exposed women with young children to higher "social costs" and may have also caused them to have second thoughts about their work aspirations and made the possibility of postponing, if not giving up, a career seem more appealing. Sofia (with one daughter aged 4 and a husband who was a freelance professional) became an associate professor in 2019, in addition to undertaking research as a freelance professional. Before the pandemic, she commuted between a city in central Italy and one in the North where her university was located. As she did not have the support of either parents or in-laws, she recounted a prepandemic life consisting of working at weekends and during holidays and antisocial hours until midnight/1 am so that she could devote around three hours (from 6 to 9 pm) to her daughter. Her "balance" was upset by the COVID-19 crisis. During the pandemic, she had to halt all her research activities so much that she was thinking of not participating in the round of National Scientific Habilitation (ASN) as a full professor. She noted the presence of very strong differences with respect to her male colleagues, who, instead, had increased their research activities and number of publications.

> The situation at the beginning [of the pandemic] was almost, let's say, I don't know how to define it, maybe nightmarish, because my little girl was at home for all those months, for about three months. . . All my work froze, from projects to monographs—everything froze. I would have liked to participate, because I was finishing the monograph, at the National Scientific Habilitation. . . when it closed

down, therefore before the definitive lockdown. . . already in the period between 20th February and 9th March. In those two weeks I did nothing and above all I threw in the towel on completing the monograph and reluctantly abandoned, after so much work, the idea of submitting an application to the National Scientific Habilitation. . . All my activities stopped. That is, I only continued with teaching— online teaching, which started immediately. . . but I did no research. . . My little girl took up all my time. But I saw instead that the lockdown had absolutely no repercussions on men. In fact my male colleagues during the lockdown, it is not that they finished [their own] book, they really wrote it from scratch, because the time, that is, having all that time available, for those who do research locked in the house is the optimum, it is the ideal, what we all dream of. . . I have many male colleagues, also with children, who have finished some works, started others, published. But this is not possible for a woman.

(Sofia, 41 years old, SSH field, ACa.)

While we find the "motherhood penalty" regardless of social class, it must be said that for those in unstable and precarious posts, given the high competition and limited opportunities for entry into academic careers, the social cost is harsher, and these second thoughts take on the appearance of true quitting. Franca, for example, felt she was playing an unfair game with her peer group and had, therefore, made several sacrifices so as not to slow down too much and still fulfill the expectations of her mentor. All considered, she was also backtracking on the idea of pursuing an academic career.

Now, during Covid, I set my alarm clock for four in the morning because I have to get on with the book I want to finish, and I work until one, eh. . . this is every day. . . I want to finish this book and I don't know how to find the time. . . In fact, I have to say that during this quarantine I had some moments of severe discouragement eh, because it has, in my opinion, widened the gap a lot. In the sense that my unmarried peers, that is, I felt that the amount of what they were writing was the incredible, right? You're locked in your home with your mom cooking you food. . .and so on. It's time you wrote anything at all. . . And I'm here having to navigate the teaching of my children, because there's that too! (Franca, 35 years old, SSH field, ECa.)

We found an emphasis on the individual responsibility of mothers in academia for both their "successes" and their "failures". In the case of men, however, the picture is more mixed. We found a group consisting of the more "innovative" fathers, those directly involved in care work, or those more sensitive to the issue of reconciling family and work. For example, Mario, whose wife was a pharmacist (and, therefore, an essential worker), stayed at home with his 6-year-old son. He said: "*I managed*". Raffaele, 40, alternated with his wife, "*even if she did more*". Mimmo, with one 2-year-old son and awaiting a second child, posed interesting reflections on the redefinition of life–time borders following the pandemic.

The first day with my son, I have this distinct memory of me [laughter], while my wife was trying to recover from the caesarean, dead tired, I was sleeping with the baby in my arms and on the other side my cell phone to reply to emails so. . . obviously no one was forcing me to stay at the emails on that day, but in the meantime I was there. (Mimmo, 33 years old, SSH field, ECa.)

Within this somewhat heterogeneous frame of fatherhood, also apparent was that of the "naturalization" of gender roles. Leo, who had a 2-year-old daughter, said in regard to possibly different difficulties affecting mothers and fathers and coherently with the culture of parenthood prevalent, above all, in the first years of children's lives that:

There are also moments in the growth of a person in which the role of a father and the role of the mother, without necessarily having to belittle one or value the other, may be somewhat different, but not because one it is more important, one

is less important, but because while I was growing up, there was a moment in which I was looking for my mother. And I see that my daughter does so as well, and so yes, sometimes there is a representation certainly with some stereotypes that are a bit eh... particular, however, it doesn't mean that an aspect is devalued or exalted, it's part of a period of life. (Leo, 33 years old, SSH field, ECa.)

There was also a group of fathers, more like the group of mothers, who recognized the negative impact of parenthood on productivity and spoke of the pleasure of being able to reinvent oneself and devote time to fatherhood:

Covid impact has had its own meaning because of course having the baby at home since February, my wife and I have taken turns. A bit we have used the babysitter in the latest period [of Covid], obviously not in the initial one, when there were no grandparents, no babysitter. Nobody, so we—I have to say her more than me—but we accordingly managed... our times. We were dad and mom [laughing]. (Raffaele, 40 years old, SSH field, ECa.)

Then, there were those interviewees, especially women, at a stage in their life course where they had to take care of nonself-sufficient family members and decide how and where to pass the lockdown, who, to not fail in this responsibility, had to leave their partner in another city:

I returned to \*\*\*\* for my father, because, I mean, between the two [the partner and the father], I necessarily had to choose my father... Last year my father had a cerebral haemorrhage despite being well, but again... one thing is the possibility of, in four hours, two hours going back to \*\*\*\* and managing it, in short, one thing is instead... a perspective that we used to see... At that moment there was a lot of fear... so my father was left alone without... at first the cafés and bars were still open, but then I thought that that would be an activity again, that is... I saw the film in advance... and I said if they close it to him... there is no caregiver [meaning the law that allowed family caregivers to move] because he is self-sufficient. So between my partner, I chose my father. (Pia, 48 years old, STEM field, ACa.)

## 6. Discussion

Through an intracategorical intersectional analysis (McCall 2005) of a large volume of qualitative data about the working lives of academics during the COVID-19 pandemic, in this article, we have attempted to complexify the analysis by shifting the focus to other axes in addition to the more commonly used ones of gender and motherhood and including class (career position) and age (individuals' life-course stage). In so doing, we found that even if more generally the COVID-19 pandemic imposed new constraints on academic work by restricting the opportunities for face-to-face encounters and requiring a large amount of time to learn online teaching methods, the unprecedented experience of the pandemic also laid the bases—to different extents according to gender, class, and individuals' life-course stage—for adherence to the model of the "ideal academic" (Cannito et al. 2023; Lund 2015); a model that entails working nonstop and without a timetable in order to increase scientific output. By focusing on the various components of academic work, this article shows how working and life activities were redefined during the pandemic, especially the lockdowns, thus producing "unprecedented" uses and abuses of time, meaning, for example, more hours devoted to teaching and online meetings, particularly for advanced career academics but also opportunities related to the use of time because remote working made it possible to "free up time for oneself" for those living in couples without caring responsibilities and "optimize time"/"gain time" in the experiences of some subjects, i.e., men and women alike who lived alone or with their parents, for whom the home became a single productive time–place.

Moreover, since our qualitative analysis did not take the three axes individually but brought out the subjective experiences at neglected multiple points of intersection,

it confirms the "motherhood penalty" found in other research (e.g., Minello 2020) and highlights different nuances of the phenomenon. The penalty in terms of work–life balance and scientific productivity regarded, above all, women with young children and daughters taking care of their elderly parents, while those who lived alone or with parents, as well as those who lived in a couple with adult children or without caring responsibilities, had very little to reconcile. Not only are gender and age (individuals' life-course stage) relevant, but class (career position) is too as it exposes people differently to the neoliberal culture of productivity and competition and to distinct work commitments. Whereas in the case of advanced career academics (professors), we observed that the pandemic forced women with caring responsibilities to postpone research in order to devote themselves to other aspects of academic work (mainly teaching and service work), for those in unstable and precarious posts, we observed that the difficulties of being as productive as the peer group and meeting the expectations of scientific mentors made young mothers consider quitting their careers.

Importantly, the disadvantage in terms of career progress for women with care responsibilities emerged not only with respect to male and female colleagues who did not have such responsibilities but also with respect to the group of fathers. The analysis found that the group of fathers was more heterogeneous in terms of the impact of the lockdown on work: only some of the fathers recounted difficulties of work–life conciliation.

The intersectional approach allowed us to show the complexity of the phenomenon and a great heterogeneity of experiences. In this regard, surprisingly, there are also those who derived various advantages from the reorganization of work during the pandemic and reaped various emotional benefits. The analysis shows that both men and women who were at a stage in their life course with a cohabiting partner but without caring responsibilities reported positive experiences of work–life balance, especially those whose living conditions allowed them to have good control over their own space–time, a condition that—as related to social class—advanced-career academics were especially able to take advantage of.

## 7. Conclusions

In a context characterized by strong gender asymmetries, even before the COVID-19 outbreak, this article on the pandemic period and the academic world in Italy shows, firstly, how in the field of gender equality, there is always a risk of backlash because progress along one line (e.g., a reduction in the imbalance in top positions) can be accompanied by regressive phenomena. Secondly, this research suggests that in order to capture such regressive phenomena on the path to gender equality, it makes sense to complexify the analysis and capture the diversity of lived experiences through the interlocking of the gender dimension with other axes, in particular, social class and age. As we detailed in the previous section, the lived experiences were very different. In several cases, both men and women with certain characteristics were able to "benefit" from the pandemic by working nonstop and adhering to the model of the "ideal academic", potentially leading to faster career advancement. Importantly, we show that the phenomenon of regression did not affect all women equally and led to different outcomes, from physical/mental exhaustion and consideration of slowing career progression among associate professors to the hypothesis of abandoning an academic career or giving up building a family among younger women in the early and precarious stages of their careers. Social class also played a role as it facilitated work–life balance and well-being at work, except for respondents who had young children, especially mothers. However, a limitation of this study is that we cannot observe the concrete consequences of this diversity in terms of individuals' academic productivity, career progression, and life choices. We think it is important for future research to address these issues by conducting longitudinal studies that examine academic workers over time and at multiple points in time.

In policy terms, to reduce gender inequalities, this study recommends that along with some structural actions (e.g., relieving those with caregiving burdens through teaching assistants or through a babysitting bonus or extra research funds), there is much room for

cultural action. First of all, university organizations should act on that "*inhuman*" model of the "ideal academic" to make it more sustainable in terms of quality of working life and work–life balance. Secondly, to erode the association between women and care work, which relegates and overburdens them to their private life, it is important to revalorize and repoliticize care work, making it more visible, even within organizations, for example, in the forms of campaigning for the recognition of so-called "academic domestic work". At the same time, in order to avoid the risk of the "motherhood penalty" and reproducing the division of gender roles in academia and society, cultural measures that de-essentialize care as feminine and discuss care work as a right and duty for all, including men, are also needed (e.g., through a campaign to promote the involvement of fathers).

**Author Contributions:** Conceptualization, A.C., M.N. and A.T.; methodology, A.C., M.N. and A.T.; software, A.C., M.N. and A.T.; validation, A.C., M.N. and A.T.; formal analysis, A.C.; investigation, A.C., M.N. and A.T.; resources, A.C., M.N. and A.T.; data curation, A.C., M.N. and A.T.; writing—original draft preparation, A.C., M.N. and A.T.; writing—review and editing, A.C., M.N. and A.T.; visualization, A.C., M.N. and A.T.; supervision, M.N.; project administration, M.N.; funding acquisition, M.N. All authors listed have made substantial field research work and intellectual contributions. A.C. wrote Sections 2 and 5.1; M.N. wrote Sections 4 and 5.3; A.T. wrote Sections 3 and 5.2. Sections 1, 6 and 7 were jointly written by all the authors. All authors have read and agreed to the published version of the manuscript.

**Funding:** This research was funded by MIUR (the Italian Ministry of Education, University, and Research). Grant Number Prot. 2017REPXXS.

**Institutional Review Board Statement:** The study was conducted in accordance with the Declaration of Helsinki, and approved by the Ethics Committees of University of Turin, established by R.D. No. 6502 dated 23 October 2008, our project (as lead) was approved by Prot. No. 191917 dated 27 May 2020.

**Informed Consent Statement:** Informed consent was obtained from all subjects involved in the study.

**Data Availability Statement:** The data presented in this study are available on request from the corresponding author. The datasets presented in this article are not readily available because research ethics and data anonymity do not allow access to the first level of interview data as they may contain information which could run the risk of identifying the participants.

**Conflicts of Interest:** The authors declare no conflicts of interest. The funders had no role in the design of the study; in the collection, analyses, or interpretation of data; in the writing of the manuscript; or in the decision to publish the results.

## Notes

[1] We are using the term "motherhood penalty" broadly, including reductions in working time, as a substantial part of the literature does in contrast with the narrower focus on wage penalties examined by Budig and England (2001) and other researchers.

[2] We acknowledge GeA project (GEndering Academia) funded by MIUR (the Italian Ministry of Education, University, and Research). Grant Number Prot. 2017REPXXS.

[3] Distance learning, especially, in the case of children attending primary school, which had relevant effects on parents' working time, was used in Italy, though to varying degrees in different regions, during COVID-19 2020–2021 for at least 5 months on average (Eurydice 2022).

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
