# Peer review of "Inequalities in Academic Work during COVID-19: The Intersection of Gender, Class, and Individuals’ Life-Course Stage"

_socsci, doi:10.3390/socsci13030162_

Round 1

Reviewer 1 Report

Comments and Suggestions for Authors

The paper is firmly grounded in the empirical literature that has emerged from the Covid-19 pandemic's effect on academics.  The intersectional perspective is thoroughly conveyed, and the choice of how it informs this empirical work well justified.  The intersectional lens of gender*life course*class is original and--as revealed by the findings--warranted. 

This paper is excellent in every way.  I struggle to find anything critical to say.  But I am an academic, so....  I think that there is an additional dimension to class not explored here (and perhaps the data would not support it).  I am convinced about the neo-liberal argument related to position type, and its Covid effects.   I think there is an additional dimension of class*life course, however, that has to do with the older age and cohort of those without caregiving responsibilities.  In short, not only are they not in precarious academic positions, but perhaps just as important:  They are very well off financially.  Perhaps this points to a generational dynamic whereby older academics, who were not subject to the neo-liberal conditions early in the career, are especially advantaged (i.e., not just lacking disadvantage).  This comment should NOT be taken as a request for revision, but perhaps for consideration as the author(s) undertake further work on this rich data set.  I thoroughly enjoyed the paper.

Author Response

Letter to the Editors and Reviewers 

January 28, 2024 

Dear Editor and Reviewers, 

Thank you for your comprehensive and constructive feedback. We are extremely grateful for your valuable advice and suggestions, which we have all endeavoured to take on board to improve the paper.  

We highlighted the substantial changes we made in the manuscript by using colored text in the MS Word file. Below we explain how we have addressed each of your concerns (our response against each point is in italics font).  

Rewiewer 1 wrote: The paper is firmly grounded in the empirical literature that has emerged from the Covid-19 pandemic's effect on academics. The intersectional perspective is thoroughly conveyed, and the choice of how it informs this empirical work well justified.  The intersectional lens of gender*life course*class is original and--as revealed by the findings--warranted. This paper is excellent in every way.  I struggle to find anything critical to say.  But I am an academic, so....  I think that there is an additional dimension to class not explored here (and perhaps the data would not support it).  I am convinced about the neoliberal argument related to position type, and its Covid effects.   I think there is an additional dimension of class*life course, however, that has to do with the older age and cohort of those without caregiving responsibilities.  In short, not only are they not in precarious academic positions, but perhaps just as important: They are very well off financially.  Perhaps this points to a generational dynamic whereby older academics, who were not subject to the neo-liberal conditions early in the career, are especially advantaged (i.e., not just lacking disadvantage).  This comment should NOT be taken as a request for revision, but perhaps for consideration as the author(s) undertake further work on this rich data set.  I thoroughly enjoyed the paper. 

Thank you very much, we are pleased that the article was appreciated. We have tried to take up this suggestion on social class*life course in the new conclusions section (section 7, added in the revised version of the article) and also, albeit briefly, in the empirical section in relation to the second household and family life course type (couples without children or with adult children), where the role of social class – in the sense of economic availability – becomes clear. This is certainly a topic that could be researched further in the future with our data set. Thank you for the suggestion! 

Reviewer 2 Report

Comments and Suggestions for Authors

this is a much needed paper, written in a very clear and fluid way. I was suprised and delighted in equal manners to finally see an investigation on the work life balance of academics from an intersectional perspective. I particularly liked that the authors observed how gender intersected with other social categories to produce specific conditions of (dis)advantage for different groups within academia during the Covid-19. 

I suggest that the authors iclude sub-titles in section 4. Context, data and method, for procedure, sample and instruments. I also need the standard deviation for age mean.

more information about the interviews. Duration, how they were conducted, etc..

I would like to see information regarding the age of children - when you mention "small children", what is the mean age of this group? This is important.

Are there any cases of single parents in your group

Author Response

Letter to the Editors and Reviewers 

January 28, 2024 

Dear Editor and Reviewers, 

Thank you for your comprehensive and constructive feedback. We are extremely grateful for your valuable advice and suggestions, which we have all endeavoured to take on board to improve the paper.  

We highlighted the substantial changes we made in the manuscript by using colored text in the MS Word file. Below we explain how we have addressed each of your concerns (our response against each point is in italics font).  

Rewiewer 2 wrote: This is a much needed paper, written in a very clear and fluid way. I was suprised and delighted in equal manners to finally see an investigation on the work life balance of academics from an intersectional perspective. I particularly liked that the authors observed how gender intersected with other social categories to produce specific conditions of (dis)advantage for different groups within academia during the Covid-19. I suggest that the authors iclude sub-titles in section 4. Context, data and method, for procedure, sample and instruments.  

Thank you so much for appreciating our work. As requested, we have added sub-titles in section 4: “4.1 The Italian academic context”; “4.2 Data and methods”. Thanks for this suggestion, we believe that the empirical section is clearer in the revised version of the article. 

Rewiewer 2 wrote: I also need the standard deviation for age mean. More information about the interviews. Duration, how they were conducted, etc.. I would like to see information regarding the age of children when you mention "small children", what is the mean age of this group? This is important. Are there any cases of single parents in your group?  

In this new version of the article, the section on methodology is much more extensive and detailed. 

Thank you for the suggestions. We have included more information about the interviews (duration, online mode, etc.), as requested. We have added information about the age of the young children (under 14) in the third household and family life course type (couples with young children and other family members in need of care), and the age of the children (over 14) in the second group (couples without children or with grown-up children). We have not included the standard deviation as it does not seem to us to be a detail that adds crucial information in a qualitative study such as ours, and we have already provided the age range of the participants for each of the two main groups in the sample (early career academics and advanced career academics). Moreover, we have clarified the sample, indicated the size of the three groups, and explained the process of analysis in a more extensive way, as requested by Reviewer 3. Finally, as suggested by the Editor, we have included information on the pandemic context in Italy as well as some results on the differences between STEM and SSH disciplines. We have no single parents in the sample. 

Reviewer 3 Report

Comments and Suggestions for Authors

Thank you for providing me with the opportunity to review the paper "Inequalities in Academic Work during Covid-19: The Intersection of Gender, Class, and Individuals’ Life-Course Stage." The study utilizes a qualitative approach, interviewing 127 Italian academics across Italy. The findings suggest that during the pandemic, certain groups of researchers and professors, depending on the intertwining of gender, position, and family situations, experienced uniquely challenging working conditions and disadvantages.

The paper is relatively easy to follow, and I appreciate the intent to investigate inequalities in the academic context using an intersectional approach. While this is an important issue and the sample is numerous and diverse, I see some challenges for the paper in its current form, mainly in the discussion section. Below are my comments for each section. Please interpret these as constructive criticism aimed at helping you bring out the full potential of your manuscript. Best of luck in further developing your interesting work.

Abstract: I suggest better synthesizing your findings and differentiating them from the aim of the study.

Introduction and research questions: I appreciate your review of gender inequalities during the pandemic and the description of the intersectional approach. Otherwise, I think it could be useful to include a preliminary brief description of the gender gap and inequalities in academia, citing relevant literature to clarify the phenomenon. This information appears only at the end of the 3rd paragraph, and some key concepts, such as the ceiling effect, are mentioned only in the method section.  This could help in better understanding the relevance of your research question in the current context.

In this regard, I suggest referring to the work of Heijstra, T. M., Einarsdóttir, Þ., Pétursdóttir, G. M., & Steinþórsdóttir, F. S. (2017). Testing the concept of academic housework in a European setting: Part of academic career-making or gendered barrier to the top? European Educational Research Journal, 16(2–3), 200–214.

Methods: I have some comments on the sample description, but most importantly regarding the description of the analysis method. As for the sample, it is not clear whether ordinary professors were included and, if not, why they were excluded. Additionally, even if it is not practicable to insert a table with the list of all the participants given the large number, I suggest considering including a table to describe your sample alongside the written text.

Regarding the analyses, the theoretical approach used in data analysis and how you came up with three groups, that you named as “ideal type,” needs clarification and further explanation.

Also, since you used gender as the first axis of analysis, I was surprised by the definition of groups based on family conditions/Life-Course Stage. This seems to be a preliminary result that needs better explanation before describing each group.

Results: Reading the results, I had some questions that could be useful to clarify. What is the size of each main group? More importantly, are there references in the data regarding relationships with colleagues and the university context and practices that influence the reproduction of inequalities? While it seems to be one of the themes of your interviews, this topic does not seem to take up much space in the results, except as a reference to the dominant performance culture in the university context that participants have internalized. Finally, what are the implications in terms of work activities and career for the second group? This aspect does not appear described as in the other two.

Discussion: The discussion section does not go much beyond the description of what you have found. I was expecting to see more conceptual development and, above all, a discussion of the relevance of the results to the current academic context. Your results help clarify what happened during a very complex time in our society, but what can they tell us about the current situation? I would suggest reasoning more about the practical implications of these results and future research directions, as well as discussing the limitations of your study.

I hope you find some helpful comments in my review. I would really like to see that manuscript published.

Author Response

Letter to the Editors and Reviewers 

January 28, 2024 

Dear Editor and Reviewers, 

Thank you for your comprehensive and constructive feedback. We are extremely grateful for your valuable advice and suggestions, which we have all endeavoured to take on board to improve the paper.  

We highlighted the substantial changes we made in the manuscript by using colored text in the MS Word file. Below we explain how we have addressed each of your concerns (our response against each point is in italics font).  

Rewiewer 3 wrote: Thank you for providing me with the opportunity to review the paper "Inequalities in Academic Work during Covid-19: The Intersection of Gender, Class, and Individuals’ Life-Course Stage." The study utilizes a qualitative approach, interviewing 127 Italian academics across Italy. The findings suggest that during the pandemic, certain groups of researchers and professors, depending on the intertwining of gender, position, and family situations, experienced uniquely challenging working conditions and disadvantages. The paper is relatively easy to follow, and I appreciate the intent to investigate inequalities in the academic context using an intersectional approach. While this is an important issue and the sample is numerous and diverse, I see some challenges for the paper in its current form, mainly in the discussion section. Below are my comments for each section. Please interpret these as constructive criticism aimed at helping you bring out the full potential of your manuscript. Best of luck in further developing your interesting work. 

Abstract: I suggest better synthesizing your findings and differentiating them from the aim of the study.  

Thanks very much for the suggestion. As requested, we have revised the abstract to include our main findings. 

Rewiewer 3 wrote: Introduction and research questions: I appreciate your review of gender inequalities during the pandemic and the description of the intersectional approach. Otherwise, I think it could be useful to include a preliminary brief description of the gender gap and inequalities in academia, citing relevant literature to clarify the phenomenon. This information appears only at the end of the 3rd paragraph, and some key concepts, such as the ceiling effect, are mentioned only in the method section.  This could help in better understanding the relevance of your research question in the current context. In this regard, I suggest referring to the work of Heijstra, T. M., Einarsdóttir, Þ., Pétursdóttir, G. M., & Steinþórsdóttir, F. S. (2017). Testing the concept of academic housework in a European setting: Part of academic career-making or gendered barrier to the top? European Educational Research Journal, 16(2–3), 200–214.  

Thank you very much for this suggestion, which we have followed to strengthen the article in this new version. In the introductory section, we have included a brief description of the phenomenon of gender inequalities in the academic context, citing three key metaphors: the leaky pipeline, the glass door and the glass ceiling. We have also included the article by Heijstra et al, 2017 you pointed out. Many thanks for this suggestion! 

Rewiewer 3 wrote: Methods: I have some comments on the sample description, but most importantly regarding the description of the analysis method. As for the sample, it is not clear whether ordinary professors were included and, if not, why they were excluded. Additionally, even if it is not practicable to insert a table with the list of all the participants given the large number, I suggest considering including a table to describe your sample alongside the written text. Regarding the analyses, the theoretical approach used in data analysis and how you came up with three groups, that you named as “ideal type,” needs clarification and further explanation. Also, since you used gender as the first axis of analysis, I was surprised by the definition of groups based on family conditions/Life-Course Stage. This seems to be a preliminary result that needs better explanation before describing each group. 

In this new version of the article, the section on methodology is much more extensive and detailed. We have clarified that the sample is made up of two groups of academics who are at different stages of their academic careers but are not full professor (yet): post-docs and temporary researchers – Early Career academics (ECa) – and associate professors – Advanced Career academics (ACa). We did not include full professors in order to observe the consequences of the pandemic on academics' career progression strategies and aspirations. As requested by Reviewer 2, we also included more information about the interviews (duration, online mode, etc.) and added information about the age of young children (under 14 years old) in the third household and family life course type and the age of children (over 14 years old) in the second group. However, we think that it would not be possible to include a table describing our large sample consisting of 127 interviews and the main sample information is already included in the methodology section. 

As far as the analysis is concerned, we have explained it in more detail. In the first phase of the analysis, a thematic analysis was carried out in several steps following an iterative process with the support of the software Atlas.ti. The analysis looked in depth at the everyday working lives and work-life balance of early and advanced career academics during the pandemic. In a second more interpretative phase, we sought to shed light on the intersections of three axes of diversity which were made more visible by the pandemic: gender, social class (precarious versus stable and prestigious career positions) and age (individuals’ life course stage). In this phase, we identified three 'conformations' that best embody the intersectionality of these axes as they show specific outcomes in terms of quality of working life, well-being, and potentially unequal long-term consequences for the careers of academics. Throughout the analysis process, the codes and their inter-relations were discussed among the authors, and a continuous conversation was maintained between the coding and theoretical interpretations. We came out with three groups. We re-elaborated a bit our outcomes, reframed in household and family life course types, instead of “ideal-types”. Finally, as suggested by the Editor, the methodological section has been integrated with information on the pandemic context in Italy as well as some results on the differences between STEM and SSH disciplines. 

Rewiewer 3 wrote: Results: Reading the results, I had some questions that could be useful to clarify. What is the size of each main group? More importantly, are there references in the data regarding relationships with colleagues and the university context and practices that influence the reproduction of inequalities? While it seems to be one of the themes of your interviews, this topic does not seem to take up much space in the results, except as a reference to the dominant performance culture in the university context that participants have internalized. Finally, what are the implications in terms of work activities and career for the second group? This aspect does not appear described as in the other two. 

As requested, we have included the size of each group in this reviewed version of the paper: living alone or with parents (32 interviewees), couples without children or with grown-up children, i.e. over 14 years old (52 interviewees), and couples with young children (under 14 years old) and other family members in need of care (43 interviewees). Thank you for the suggestion. As for data on the organizational context in a narrower sense (relationships between colleagues, gendered practices within the organization, etc.), these are present in our dataset but are the subject of other publications that do not focus on the period of the pandemic. Finally, we have written more explicitly about the implications in terms of work activities for the second group. Thank you for this suggestion. As in the case of the first household type, a culture of overwork – pre-existing but amplified by the pandemic – emerged in couples without children or with adult children without distinction of gender and both in the STEM and in SSH.  Compared to those living alone, however, we did not find a feeling of being completely absorbed in the work sphere and alienated from the social world. The interviewees in the second group worked long hours, but they did so while reconciling with their private lives.  

Rewiewer 3 wrote: Discussion: The discussion section does not go much beyond the description of what you have found. I was expecting to see more conceptual development and, above all, a discussion of the relevance of the results to the current academic context. Your results help clarify what happened during a very complex time in our society, but what can they tell us about the current situation? I would suggest reasoning more about the practical implications of these results and future research directions, as well as discussing the limitations of your study. I hope you find some helpful comments in my review. I would really like to see that manuscript published. 

Thanks for this helpful comment. In the revised version of the article, section 6 of the conclusions has become the discussion section, while we have added a new conclusions section. In this new concluding section, we have explicitly written what in our opinion, is the relevance of our findings in the current academic context. Firstly, our findings show how in the field of gender equality there is always a risk of backlash, because progress along one line (e.g. a reduction in the imbalance in top positions) can be accompanied by regressive phenomena. Secondly, this research suggests that in order to capture such regressive phenomena on the path to gender equality, it makes sense to complexify the analysis and capture the diversity of lived experiences through the interlocking of the gender dimension with other axes, in particular social class and age. Also, as suggested, we have clarified a limitation of our research in this regard: precisely, we cannot observe the concrete consequences of this diversity in terms of individuals' academic productivity, career progression and life choices. We think it is important for future research to address these issues by conducting longitudinal studies which examine academic workers over time and at multiple points in time. Finally, as requested by the Editor, in this last section we have reflected on the possible implications for organisational policies and cultural actions that could be taken in the university context. 

Round 2

Reviewer 3 Report

Comments and Suggestions for Authors

I appreciate your efforts in responding to the comments, and I believe the work can be published in its present form.

Author Response

Dear Reviewer, 

Thank you for your comprehensive and constructive feedback. We are extremely grateful for your valuable review.

Best regards